# Augmenting Transformers with KNN-Based Composite Memory

## Abstract

Various machine learning tasks can benefit from access to external information of different modalities, such as text and images. Recent work has focused on learning architectures with large memories capable of storing this knowledge. We propose augmenting generative Transformer neural networks with KNN-based Information Fetching (KIF) modules. Each KIF module learns a read operation to access fixed external knowledge. We apply these modules to generative dialogue modeling, a challenging task where information must be flexibly retrieved and incorporated to maintain the topic and flow of conversation. We demonstrate the effectiveness of our approach by identifying relevant knowledge from Wikipedia, images, and human-written dialogue utterances, and show that leveraging this retrieved information improves model performance, measured by automatic and human evaluation.

## 1 Introduction

Machine learning solutions to various tasks, such as game-playing or dialogue, are often dependent on external information. This information can take multi-modal forms, including structured knowledge bases, free text, and images, and also comes in overwhelmingly large quantities. A pressing challenge is to create models that can identify which specific elements of multiple information sources are relevant, and incorporate them into standard architectures on each task.

Previous work has explored incorporating large external memories into neural network layers (Weston et al., 2014; Sukhbaatar et al., 2015; 2019; Lample et al., 2019). Many existing approaches focus on using attention over the memory slots, which is computationally intensive and becomes less effective as the the size of the memory grows. In this work, we propose representing multiple sources of external information as fixed encodings and using K Nearest Neighbors search to fetch relevant information. KNN search is computationally efficient and scalable, and libraries like `faiss` (Johnson et al., 2019) allow KNN to be easily used on GPUs and integrated into neural networks. As the external memories are kept fixed, they do not require any training to learn the memories along with the model. We can thus scale more easily to larger memories by learning only the KNN-based read operation to identify relevant information from the memory.

Our core contribution proposes an efficient, KNN-based Information Fetching (*KIF*) module that can access relevant external knowledge, combine knowledge from different sources, and integrate this information into standard sequence to sequence architectures. We apply these flexible modules to two dialogue datasets, challenging tasks where generative models can leverage external information to write coherent, on-topic responses. We show that relevant information can be identified from hundreds of thousands of candidates in a multi-modal, multi-knowledge-source setting to improve the performance of generative dialogue models. On both datasets, we achieve state of the art results compared to generative models and match the quality of retrieval models.

## 2 Related Work

**Incorporating External Knowledge into Neural Networks.** Augmenting neural networks with memory, or longer term components that can be accessed with read and write operations, has been

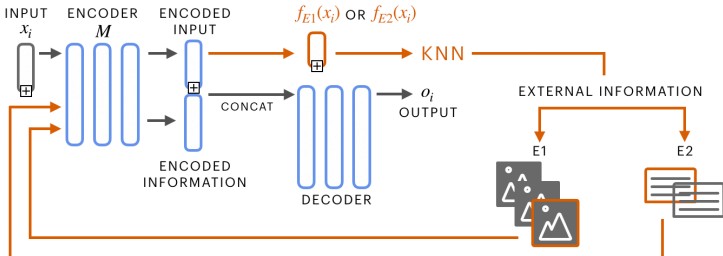

Figure 1: **KIF** modules (orange) fetch relevant information from multi-modal external knowledge sources and incorporate it in standard neural architectures.

explored in various proposed architectures. For example, Memory Networks (Weston et al., 2014; Sukhbaatar et al., 2015; 2019) introduce attention mechanisms over large external memories. Neural cache models (Grave et al., 2016) simplify these to access previous memories with a dot product. Previous work has studied how to read and write into these memory architectures (Rae et al., 2016; Graves et al., 2014; Joulin & Mikolov, 2015). Another line of research has focused on computational scalability for larger external memories. For example, Chandar et al. (2016) propose a hierarchical memory network rather than a flat one and Rae et al. (2016) learn sparse operations to read and write. Lample et al. (2019) focus on learning memories of up to one million slots and how to efficiently access the slots using product keys. Beyond explicit memory representations, it may be possible to store information implicitly during training time by memorizing common patterns present in text (Petroni et al., 2019). We focus on learning to fetch relevant information from multiple explicit external multi-modal knowledge sources and integrate them into one network.

Work has also focused on computationally efficient softmax operations (Mnih & Hinton, 2009; Grave et al., 2017; Chen et al., 2015). Many approximate softmaxes use KNN-like operations to form clusters, and the softmax operation is constrained by the slow calculation of the exponential. Our use KNN benefits from efficient and scalable libraries such as `faiss` and `nmslib`.

**Generative Dialogue.** We develop a general architecture for incorporating external information and apply it to the case of generative dialogue models. Previous work in dialogue has leveraged knowledge as necessary information to accomplish the task. For example, airline and restaurant booking tasks often use API calls to access information about reservation times and availability (Bordes et al., 2016). In contrast, our work focuses on how to incorporate unstructured knowledge, such as free text found on the web. Previous work has employed architectures that attend over the available knowledge and identify relevant pieces of information, which scales poorly with large quantities of information (Dinan et al., 2018; Qin et al., 2019; Lian et al., 2019). In this work, we replace the use of attention over external information with the output of our KNN module.

On the modeling side, work has explored both generative (Serban et al., 2016a;b) and retrieval based models (Zhang et al., 2018), which identify the best utterance from the training set to return as the dialogue response. This often leverages self-attention or cross-attention mechanisms (Humeau et al., 2019). Further work has explored hybrid models, for example using the output of a retrieval model as input for a generative model (Dinan et al., 2018; Weston et al., 2018). We extend these approaches by augmenting generative models with retrieval-like operations based on KNN search, allowing dialogue models to flexibly incorporate various sources of external knowledge.

## 3 KNN-BASED INFORMATION FETCHING MODULES

Broadly, the KNN-based Information Fetching (KIF) module assumes a model $M$ can access inputs $X = x_1, x_2, \ldots, x_n$ to produce outputs $O = o_1, o_2, \ldots, o_n$. In a setting without additional supporting information, the model will process inputs to make output predictions: $M(x_i) = \hat{o}_i$. However, in many tasks, additional information is present, represented as $E = \{e_1, e_2, \ldots, e_m\}$. To incorporate $E$ into $M$, we encode each element of $X$ and $E$ to a fixed-size vector representation. This can be accomplished in a variety of ways, for example with an encoder neural network.

Then, to make predictions, the model encodes $x_i$ and uses K Nearest Neighbors to find the closest related information in $E$. The representations of the identified nearest neighbors are combined in a weighted sum, where each of the $k$ retrieved neighbors is weighted by its similarity to $x_i$.

These operations are differentiable, so they can be incorporated into neural networks in a straightforward way. All elements of the knowledge source $E$ are pre-computed and kept fixed — we do not backpropagate to affect the embeddings of the pre-encoded knowledge. However, this lack of backpropagation can introduce a mismatch between the encoding of $E$ and the model that is training, as the training model has constantly changing representations because the weights are being learned. The model must learn a function to align its representations to the external memory. To circumvent this misalignment, we learn a mapping operator $f_E(x)$ that maps elements of $X$ into the information representation space $E$. Concretely, $f_E(x)$ is a multi-layer perceptron with ReLU nonlinearities. From the input elements of $X$, $f_E(x)$ learns a representation of an output close to the corresponding projection of $X$ into $E$. This can be interpreted as learning a read operation on a fixed external memory. If there was no change to the encoding of the model compared to the pre-computed knowledge, then the ideal mapping operator would be the identity function. However, as the model changes significantly during the training process, the nonlinear mapping capability of $f_E(x)$ is essential to be able to identify the correct knowledge $E$ from the input $X$.

Thus, a model augmented with KIF will incorporate external knowledge in the following manner. First, we find the $k$ nearest elements to the projection of $x_i$ in $E$ based on KNN search using inner product, and then the relevant elements are encoded by $M$.

$$\mathtt{KIF}_i = \Big\{ M(e) \mid e \in \mathtt{KNearest}\big(E, k, f_E(x_i)\big) \Big\} \tag{1}$$

These elements are weighted by their nearest neighbor scores and summed. This is then concatenated to the representation of $x_i$ and used by $M$ to form the prediction:

$$M([x_i, \mathtt{WeightedSum}(\mathtt{KIF}_i)]) = \hat{o}_i \tag{2}$$

This is easily extended to using multiple modules simultaneously. For instance, two sources of information, $E_1$ and $E_2$, can be combined by identifying the top candidates of each information source. The weighted sum of the KIF output on each information source is concatenated with $x_i$.

Finally, different sources of information may not be required for every prediction and some information sources can be more important than others. To allow the model to make more fine-grained decisions about what information to use from what source, and how much of it, we add a gating mechanism using a sigmoid function around each weighted sum of KNN representations. $\mathtt{KIF1}_i$ and $\mathtt{KIF2}_i$ denote the KIF module from Equation 1 applied to $E_1$ and $E_2$ respectively.

$$\mathtt{WS1}_i = \mathtt{WeightedSum}(\mathtt{KIF1}_i) \tag{3}$$
$$\mathtt{WS2}_i = \mathtt{WeightedSum}(\mathtt{KIF2}_i) \tag{4}$$
$$M\big([x_i, \ \sigma(\mathtt{WS1}_i) \cdot \mathtt{WS1}_i, \ \sigma(\mathtt{WS2}_i) \cdot \mathtt{WS2}_i]\big) = \hat{o}_i \tag{5}$$

## 4 Applying KIF to Dialogue Tasks

We describe how to apply our method to the task of generative dialogue, a challenging setting where models must autoregressively generate engaging and on-topic responses. We investigate dialogue for two main reasons: first, dialogue agents must be able to consult relevant information to maintain the topic of the conversation. Second, retrieval-based agents have strong performance compared to generative ones, due to their ability to copy dialogue utterances from the training set. Using KIF, we can incorporate the benefits of retrieval architectures into generative, knowledge-based models.

**KIF for Generative Dialogue** In a dialogue setting, $x_i$ represents the text of the conversation $i$. A conversation consists of multiple back-and-forth *utterances* (or turns). For example, a conversation could consist of 4 turns: $x_i = [x_{i,1}, x_{i,2}, x_{i,3}, x_{i,4}]$ where $x_{i,4}$ is the direct utterance the model should respond to, and the earlier utterances are the *conversation context*.

Standard generative dialog models use a Transformer neural network as $M$ and want to produce an output $o_i$ that is an appropriate response to the conversation. However, in many cases, the

conversation history alone does not include all of the information required to produce an appropriate response. To incorporate knowledge, models often concatenate a knowledge source $E$ such as Wikipedia to $x_i$, such that $M([x_i, e_1, e_2, \ldots, e_n]) = \hat{o}_i$, and use attention modules to identify the most relevant knowledge. However, this approach is computationally intensive when handling large quantities of information. Further, attention mechanisms have been found to operate poorly over long sequences, as the mechanism is blurry and struggles to make fine-grained decisions (Fan et al., 2018). The same is true for hierarchical approaches, which lack scalability.

We augment Transformer sequence-to-sequence (seq2seq) networks with KIF. We experiment on two dialogue tasks, Wizard of Wikipedia (Dinan et al., 2018) and Engaging Imagechat (Shuster et al., 2018). We use `faiss` (Johnson et al., 2019) to perform KNN search efficiently and at scale.

**Wizard of Wikipedia**  The goal of the Wizard of Wikipedia dataset is to train knowledgeable agents that can chat in any domain. The dataset contains 1,365 various topics discussed in 18,430 dialogues in the training set, totalling 166,787 training utterances. The topic is included as the first utterance of the conversation. The dataset includes relevant Wikipedia sentences for each turn of the chat, identified by an information retrieval system.

Our model for Wizard of Wikipedia has access to two sources of external information, $E_1$ and $E_2$:

- $E1$ *is Wikipedia Knowledge* provided by the dataset as evidence to support knowledgeable chitchat. The scale of this KNN search is to filter through an average of 34 sentences. The KIF module uses dialogue features to fetch relevant knowledge to condition upon to generate the subsequent utterance.
- $E2$ *is Training Utterances*. To incorporate the benefits of retrieval-based dialogue models to the generative setting, we use KIF to identify relevant utterances from the training set and take their *responses* as input. If many conversations about dogs have already occurred, models should be able to take advantage of these human-written examples to improve their generations. There are around 170K dialogue utterances as inputs to KNN search. This can be interpreted as incorporating the benefits of retrieval models by identifying an utterance with similar structure as the text the model would like to generate.

Access to these two sources of knowledge can be seen as as learning a template and a topic separately. Sample templates can be identified from the training utterances, and topic-specific information learned by accessing the Wikipedia knowledge.

To better identify relevant training utterances from the large quantity available, we break down $x_i$ into conversation sub-features for a more fine-grained match in the KNN search step. We concatenate the encoding of the most recent dialogue utterance (e.g. $x_{i,\text{last}}$) with the encoding of the dialogue context and the turn number. These are known to be salient conversation features. The most recent dialogue utterance is the direct turn the model is responding to, and the dialogue context may provide additional clues. The turn number is important, as earlier turns are often more generic (e.g. *hi, how are you doing today*) and later turns are more specific.

**Engaging ImageChat**  The goal of Engaging ImageChat is to create agents capable of chitchatting about images, selected from the YFFC100M dataset (Thomee et al., 2015). The dataset contains 186,782 dialogues in the training set, each about a unique image, totalling 355,862 utterances. Agents are assigned one of 215 personalities (e.g. *sweet*) to increase engagingness. We use a Multi-Modal neural network designed to handle both image input and text input. Following Shuster et al. (2018), the images are encoded using a ResNeXt (Xie et al., 2017). To extract the final image representation, we project the 2048-dimensional output of the image encoder to 512-dimensions using a deep multi-layer perceptron with ReLU activation units. The conversation history, which includes the personality, is encoded with a Transformer encoder network. The image and conversation are combined using the Multimodal Sum Combiner proposed in Shuster et al. (2018).

Our model for Engaging Imagechat has access to two sources of external information, $E_1$ and $E_2$:

- $E1$ *is Chat on Similar Images*. While there are over 180K different images used in this dataset, many of the images are similar. For example, conversations associated with two pictures of dogs could be relevant to each other. The model is able to fetch from around 160K different images and returns 6 turns of related chat for each image. Fetching from $E_1$

| Model | F1 (Seen) | F1 (Unseen) |
|---|---|---|
| Retrieval Trans. MemNet* | 15.4 | 12.4 |
| 2-Stage Generative MemNet* | 18.9 | 17.4 |
| Generative Trans. MemNet* | 16.9 | 14.4 |
| + Reddit Pre-Train | 17.6 | 16.3 |
| Retrieve and Refine | 18.2 | 17.9 |
| KIF-Augmented Transformer | **26.9** | **23.3** |

Table 1: Results on the **Wizard of Wikipedia** dataset. * denotes results from Dinan et al. (2018)

| Model | F1 |
|---|---|
| Retrieval Trans.* | 9.8[1] |
| Generative Trans. MemNet | 7.1 |
| + Reddit Pre-Train | 12.8 |
| Retrieve and Refine | 13.6 |
| KIF-Augmented Transformer | **14.4** |

Table 2: Results on the **Engaging Imagechat** dataset. * denotes results from Shuster et al. (2018)

consists of identifying Related Image Chats, or conversations on similar topics (as similar images are likely to have similar conversations).

- *E2 is Training Utterances.* Similar to the motivation for the previous dataset, we allow the model to identify training utterances that could be useful for responding in the current conversation. The scale of this fetching task is large: 350K dialogue utterances. This could be interpreted as identifying utterances with similar structure to what the model would like to generate, and is complementary to the topic-based Related Image Chats.

To identify relevant information from the training utterances, we use the same dialogue features in the KNN search step, with one modification: we add the *personality* provided by the dataset. As utterances from speakers with the same personality are likely to be more related, this feature improves the quality of the fetched information.

## 5 EXPERIMENTAL SETUP

### 5.1 IMPLEMENTATION DETAILS

We use `parl.ai` (Miller et al., 2017) to implement our models. We use byte-pair encoding (Sennrich et al., 2015) to represent the text to better handle the rare word problem (Dinan et al., 2018; Fan et al., 2017). Our generative Transformer models have 8 encoder layers and 8 decoder layers, with FFN size 2048, embedding dimension 512, and 4 attention heads. We optimize using Adam (Kingma & Ba, 2014) and the inverse square root learning schedule (Vaswani et al., 2017) with 10k warmup updates. The initial learning rate is 0.0001 and we optimize for model perplexity. We use a dropout of 0.5 and set gradient clipping to 0.1. We set k = 5 for all cases. We pre-train the Transformer seq2seq model used for both datasets on 250M comments from Reddit. The comments are parsed to maintain conversational threads, so the encoder network has been exposed to conversational context at training time. The ResNeXt encoder is pretrained on 3.5 billion images (Mahajan et al., 2018). For both datasets, we model a vocabulary size of 54944 based on the BPE-based vocabulary from the Reddit pretraining. We tuned the learning rate and batchsize hyperparameters together. The model size is not tuned, as it was pre-trained with this size and thus kept fixed.

### 5.2 EVALUATION

**Generation** We generate with beam search, setting the beam size to 4 with 3-gram blocking.

**Automatic Metrics** Following Dinan et al. (2018), we compute *F1*, a metric of unigram overlap, between the generated utterance and the human-written utterance. For generative models, utterances are generated using beam search. For retrieval models, the next utterance is predicted by ranking the entire set of training utterances, and the highest scoring utterance is chosen.

---

[1]In Shuster et al. (2018), retrieval Transformer models report Hits@N using a fixed candidate set of 99 distractor candidates and 1 true candidate. We compute F1 using their open-sourced model by ranking the entire training set of over 350K utterances.

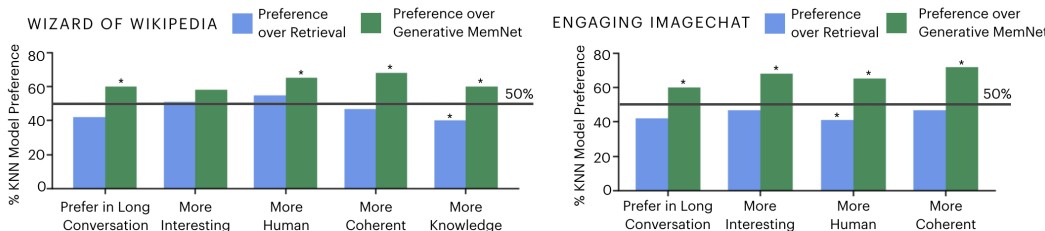

Figure 2: Human Evaluation. More than 50% indicates the KNN Model is preferred. Stars indicate statistical significance at $p < 0.05$

In Wizard of Wikipedia, there are two test sets: one set of *seen* topics, or topics that have been seen at training time with new test-time dialogues. The second set is *unseen*, or topics that have not been encountered at all during training time. We evaluate on both of these subsets.

**Human Evaluation**  We follow the setup and use the analysis questions proposed in the `Acute-Eval` dialogue evaluation system (Li et al., 2019). For reproducibility, we adopt this existing evaluation setting that has been applied to several dialogue datasets. We collect 100 human-bot dialogues on a crowdsourcing platform for both datasets. Then, we show pairs of dialogues side by side, and ask the following questions:

- Who would you prefer to talk to for a long conversation?
- If you had to say one of the speakers is interesting and one is boring, who would you say is more interesting?
- Which speaker sounds more human?
- Which speaker has more coherent responses in the conversation?
- If you had to say that one speaker is more knowledgeable and one is more ignorant, who is more knowledgeable? (Wizard of Wikipedia only)

We measure the percentage of time one model was chosen over the other, taking the agreement between three evaluators. To reduce variance, the dialogues that are paired in the evaluation were collected on the same topic for Wizard of Wikipedia and collected on the same image for Engaging ImageChat. Each topic and image used is unique and taken from the test set randomly.

## 5.3  BASELINES

We compare Transformers augmented with KIF to the state of the art retrieval models published on each dataset, as well as two additional generative baselines that have access to knowledge:

- *Transformer Memory Networks*. To contrast the ability of KIF to existing work, we compare our models to published Transformer Memory Networks (Dinan et al., 2018). These models encode each piece of external information independently with a Transformer Encoder, and these are stored as memory slots. To access information in the memory slots, a model performs dot-product attention between the memory slots and the dialogue context. In Dinan et al. (2018), the knowledge selection from Wikipedia was supervised with either a two-stage model where the first model was trained to predict the right knowledge, or an end-to-end model with an auxiliary loss for knowledge prediction accuracy.
- *Retrieve and Refine*. We implement a hybrid model (Weston et al., 2018) that incorporates top retrieval candidates as additional input to Generative Transformer MemNets.

All of the generative baselines are initialized with the same pre-training on Reddit that we use for our models for fair comparison on modeling quality. For existing published models, we re-train the open-sourced generative models with the same pre-training.

## 6   RESULTS

We describe the results of incorporating KIF modules into Transformer networks. We display an example conversation between a human and our model in Figure 4, and show the top scoring Wikipedia knowledge and Training Utterance fetched by KIF modules.

### 6.1   KIF IS EFFECTIVE FOR INCORPORATING KNOWLEDGE

**Automatic Evaluation.**   Comparing KIF augmented Transformer networks to published baselines and Retrieve and Refine, we find improved results (see Table 1 and Table 2).  For Wizard of Wikipedia, the improvement in F1 score is almost 10 points.  A major contributing factor is the construction of the dataset — as each dialogue turn is grounded in a specific knowledge sentence from Wikipedia, improving the ability to identify the relevant fact strongly improves performance. Contrasting the results from the *seen* and *unseen* test sets, the improvement on *unseen* is worse — it is harder to fetch training utterances for unseen topics. Imagechat has no explicit dependency on knowledge. We see a 2 point improvement, indicating that KIF can be generally useful.

**Human Evaluation.** Results are shown in Figure 2. On both datasets, we find there is large improvement over existing generative models (green). Evaluators agree that KIF-augmented Transformers are generally more coherent and human-sounding. Comparison to existing retrieval models (blue) is more nuanced. Along the lines of existing work (Zhang et al., 2018; Dinan et al., 2018), we find that retrieval-based models score well in human evaluations that ask how human or interesting a dialogue sounds, as they return human-written utterances from the training set.

A surprising result is that KIF-augmented Transformers are voted more human sounding than retrieval models on Wizard of Wikipedia. This is because the dataset's human utterances are long and factual due to the tendency of crowdworkers to copy Wikipedia. Sometimes humans chatting with the retrieval bot would respond *uh... that's an interesting fact?* Otherwise, our model scores similarly to retrieval models, with most of the evaluations not having statistically significant differences.

On Engaging ImageChat, while our model has significantly improved over the generative baseline, it does not beat retrieval based methods in sounding more human or being more interesting. The retrieval baseline directly copies perfectly human-written utterances from the training set, so it is a difficult baseline to beat with a generative model.

### 6.2   SCALING KIF TO CHALLENGING RETRIEVAL SETTINGS

KIF modules can be used in more realistic and challenging settings for knowledge retrieval that test the scalability of the module. In Figure 3(a), we compare the Generative Transformer MemNet Baseline with KIF-Augmented Transformers in three settings. The first is the standard Wikipedia sentences provided by the dataset (average 34 sentences). Then, we extend to providing the full Wikipedia article (average 215 sentences) and finally to providing multiple Wikipedia articles (average 1035 sentences), identified using the conversation's topic. This increasing size of available knowledge could be realistic for settings where it is unclear what information is most relevant, if filtering steps to preprocess the data remove potentially relevant information, or if information synthesis from multiple knowledge sources is necessary to produce a high quality generation. As the Wikipedia knowledge becomes more difficult to identify, performance decreases, but still outperforms the baseline that uses the dataset-provided set of 34 sentences.

Comparing the scaling capability of KIF to the standard Generative Transformer MemNet Baseline highlights the advantage of using KNN. The attention-based mechanism used in Dinan et al. (2018) struggles to identify salient information when given increasingly larger quantities of knowledge, unlike the KNN based information fetch. We hypothesize the attention mechanism is challenged by the softmax over a larger quantity of inputs, as it can be difficult to make sharp distinctions.

### 6.3   ABLATIONS

**Multiple Knowledge Sources.**   For both Wizard of Wikipedia and Engaging ImageChat, multiple knowledge sources are used — training set utterances to capture the capability of a retrieval-based model as well as knowledge from Wikipedia or related chats based on image features. The perfor-

| Model | Test F1 |
|---|---|
| *Wizard of Wikipedia* | |
| Post-Editing Seq2Seq | 17.4 |
| Retrieval Input Seq2Seq | 11.8 |
| KIF-Augmented Transformer | 18.1 |
| *Engaging ImageChat* | |
| Post-Editing Seq2Seq | 13.1 |
| Retrieval Input Seq2Seq | 10.2 |
| KIF-Augmented Transformer | 13.9 |

Table 3: Improving generation using KIF on training utterances compared to other improvements in generative models.

| Model | Valid F1 |
|---|---|
| *Wizard of Wikipedia* | |
| Previous Utterance Only | 24.6 |
| + Dialogue Context | 26.4 |
| + Turn Embedding | 27.4 |
| *Engaging ImageChat* | |
| Previous Utterance Only | 13.3 |
| + Dialogue Context | 14.5 |
| + Turn Embedding + Personality | 15.1 |

Table 4: Important Features for KNN Search using KIF. Salient conversation features improve performance on both datasets.

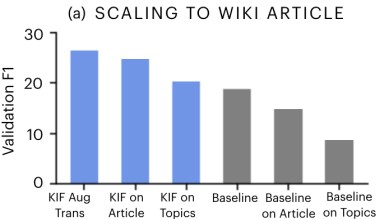 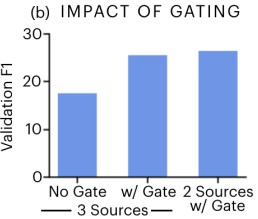 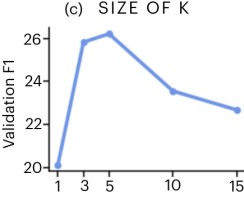

Figure 3: Ablations on Wizard of Wikipedia. (a) KIF can scale to thousands of relevant sentences (blue) while the baseline model scales poorly (gray) (b) Gating can remove irrelevant information. In the 3 Sources case, one source of external information is unrelated. (c) Performance as $k$ varies.

mance using fetching the training utterances alone with KIF is shown in Table 3. There is a decrease in performance if only the training utterances can be accessed as an external knowledge source.

We further analyze the importance of accessing training utterances compared to other possible modeling techniques to improve Transformer models. Table 3 contrasts KIF on Training Utterances with two methods of improving generative models: post-editing and using retrieved human utterances as input. Both models are Transformer seq2seq models initialized with our Reddit pre-training where the target is the human written utterance. In post-editing, the input is the predicted utterance from the best generative model. In retrieved human utterances, the input is the utterance returned by the best retrieval model. In both comparisons, we see our model is able to leverage the training set utterances returned by KIF to improve performance.

**Multiple Modalities for Fetching.** On Engaging ImageChat, we identify conversations on similar images and utterances in similar dialogue contexts. We examine the importance of being able to fetch based on the image features to access chats from related images, as the KIF module can be used for multiple modalities. Table 3 shows results without image-based KIF, which loses 0.5 F1 compared to Table 2. The large size of the training set allows separate KIF modules to be helpful. KIF using images could be interpreted as identifying topic-based information, and KIF using conversations as identifying templates for the generated response.

**Dialogue Features for KNN Performance.** The quality of the KNN search is critical to the performance of KIF modules. As the external knowledge is kept fixed, KIF must be able to align the dialogue context with the knowledge to identify relevant pieces of information. In Table 4, we show that matching on more features can improve the quality of the retrieved information. Using only the encoding of the immediate previous utterance can improve results on Wizard of Wikipedia by 7 F1 points, but this is further improved by also leveraging the encoding of context (+1.8 F1) and additionally using the dialogue turn number (+1 F1). These features are available in the datasets, and we leverage them to improve the relatedness of retrieved knowledge.

| | |
|---|---|
| **Human:** | I love disney movies! Do you like Disney? |
| **Model:** | I love disney movies, I love the fact that they have a variety of animations and animations that make them so unique. |
| | **Fetched Knowledge:** *disney consumer products and interactive media - the company was formed as a result of the merger between disney consumer products and disney interactive* |
| | **Fetched Training Utterance:** *i love going to music shows and dancing and having fun* |
| **Human:** | What's your favorite disney movie? There are so many good ones. |
| **Model:** | I love the incredibles, they are my favorite disney movie. |
| | **Fetched Knowledge:** *disney announced intentions to develop additional superhero films after the success of the incredibles* |
| | **Fetched Training Utterance:** *kiteboarding sounds very fun to me, but i have never done it* |
| **Human:** | omg, me too! the second one just came out i think. I don't watch a lot of sequels though, I think the last one I saw was toy story. |
| **Model:** | I remember when toy story was first released in 1995. |
| | **Fetched Knowledge:** *talk of a second film following toy story began around a month after the film's opening, in december 1995* |
| | **Fetched Training Utterance:** *That's neat! I like so many of her songs!* |
| **Human:** | I've actually never seen toy story before. Kind of embarrassing... |
| **Model:** | Oh... toy story is a classic! It was first released in 1999. |
| | **Fetched Knowledge:** *toy story 2 is a 1999 american computer-animated comedy film produced by pixar animation studios for walt disney pictures* |
| | **Fetched Training Utterance:** *Ohhh I see. What do people with color blindness do to cope with the effects?* |

Figure 4: Conversation between Human and KIF-Augmented Transformer on Wizard of Wikipedia. The top-scoring Wikipedia knowledge and training utterances are displayed with generated output.

**Effect of Gating.** We analyze the effect of the gating mechanism used in KIF by evaluating the capability of the gate to identify and focus on salient information. On Wizard of Wikipedia, we concatenate a third source of information: dialogue turns from a completely different corpus called PersonaChat (Zhang et al., 2018). This dataset looks quite different — short utterances without factual knowledge — and should be easy for the model to identify as distinct from Wizard of Wikipedia. As shown in Figure 3(b), if KIF on PersonaChat is included without gating, it has a harmful effect as the model includes irrelevant information. When equipped with gating, the model learns to use the gate to ignore some inputs, and can recover almost the full performance of the model without this irrelvant information source.

**Size of k.** Figure 3(c) shows the performance on Wizard of Wikipedia when varying the amount of knowledge from both sources. Generally, being able to access multiple relevant pieces of information is helpful, but too much information can be harmful. This is likely because the weighted sum operation becomes more blurry if too many sentences are summed.

## 7 CONCLUSION

We present a KNN-based Information Fetching module that learns to identify relevant information from external knowledge sources by learning a mapping-based read operation. KIF modules benefit from the scalability and efficiency of K Nearest Neighbors search, enabling computation with large external memories. We show in the context of two dialogue datasets that relevant knowledge can be identified and incorporated to create more engaging, high quality dialogue.

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

## A   APPENDIX

| **Example:** Wizard of Wikipedia |
|:---|

| | |
|---:|:---|
| **Human:** | Hey, how are you doing |
| **Fetched Training Utterances:** | *I'm great, thanks for asking. Craving some chocolate. Do you like chocolate?* |
| | *Hello, how is it going? I know some trivia about this movie* |
| **Human:** | What are your hobbies? |
| **Fetched Training Utterances:** | *I work at an elementary school. I hope you find a job you love too [...]* |
| | *I have a hound, we just got her. Although, I grew up with Labrador Retrievers.* |
| **Human:** | hi buddy, what do you think about cinematography? |
| **Fetched Training Utterances:** | *typically, a lens is used to repeatedly focus the light reflected from objects [...]* |
| | *the modern photographic camera evolved from the camera obscura* |
| **Human:** | Speaking of blue skies, have you seen the 1946 movie staring bing crosby? |
| **Fetched Knowledge:** | *blue skies is a 1946 american musical comedy film [...] and starring bing crosby [...]* |
| | *blue skies the band has since broken up* |

Figure 5: Examples of Top-2 Fetched Training Utterances and Fetched Knowledge when responding to a human chat from the dataset using a trained Wizard of Wikipedia model. Examples are taken from validation.

| Model | Valid F1 |
|:---|:---|
| KIF with Concatenation | 27.4 |
| with Addition | 18.2 |
| with Inner Product | 20.3 |

Table 5: Comparison of KIF Module Construction on Wizard of Wikipedia. KIF concatenates the fetched knowledge, which performs better compared to addition or inner product.

| Model | Valid F1 |
|:---|:---|
| *Wizard of Wikipedia* | |
| Transformer Gen MemNet | 17.6 |
| + Turn Number | 17.8 |
| + Double Last Turn | 17.8 |
| *Engaging ImageChat* | |
| Transformer Generator MemNet | 10.4 |
| No Personality Feature | 9.9 |
| Personality in Dialogue History | 10.2 |

Table 6: Comparison of Features for the Baseline Transformer Generative MemNets on Wizard of Wikipedia and Engaging ImageChat.

### A.1   ABLATIONS

**Concatenation of Fetched Knowledge.**   The knowledge of multiple KIF modules is combined by concatenating to the input representation (e.g. the dialogue context). We examine two other alternatives to incorporating knowledge from multiple sources, specifically addition and inner product. Results shown in Table 5 display that concatenation outperforms both of these alternatives, and that inner product performs better than addition.

**Adding Features Used in KIF to Baselines.**   To improve KNN search of relevant information, the KIF modules use additional features such as the dialogue turn to identify the most useful elements from the fixed memory. We investigate the importance of those features on the dialogue modeling itself by augmenting the baselines with these fetching features. Results are shown in Table 6.

For Wizard of Wikipedia, we add an explicit representation of the turn number and the concatenate the most recent turn again (as the KIF modules use these features to fetch knowledge). The addition of these features to the baseline model does not significantly improve performance.

For ImageChat, we experiment with representing the personality within the dialogue history as the KIF module uses the personality feature. In our Transformer baseline model following Shuster et al. (2018), the personality is represented as a separate feature. The removal of the personality feature is harmful, but including the personality in the dialogue history as the KIF modules do is not as good compared to the separate personality representation learned by Shuster et al. (2018).

Figure 6: (left) **Human Evaluation on the Unseen Test set** of Wizard of Wikipedia. More than 50% indicates the KNN Model is preferred. Stars indicate statistical significance at $p < 0.05$ (right) **Variance of Human Evaluation Study** on Wizard of Wikipedia. Analysis shows that multiple trials of the same experiments are relatively stable.

## A.2   ADDITIONAL HUMAN EVALUATION

Additional human evaluation results are shown in Figure 6. On the left, the same human evaluation on the Seen test set is repeated on the Unseen test set, showing similar trends.

On the right, we display an analysis of the variance of our human evaluation. The study was repeated on three different days. There is greater variance on the *More Human* and *More Interesting* questions, as perhaps different evaluators have different understanding of these aspects. Further, comparison with the Retrieval baseline has less variance compared to Generative models. It is possible that the Retrieval model is a bit easier to evaluate given the written text is always copied from a human written utterance and is usually devoid of mistakes.

