# OpenReview forum: "Augmenting Transformers with KNN-Based Composite Memory"
_ICLR.cc/2020/Conference — Reject_

### Official Review · AnonReviewer3 · 2019-10-22
**Official Blind Review #3**

**Rating:** 6

**Review:**

Summary:
- The paper proposes augmenting transformer neural networks with KNN-based information fetching modules that can access relevant external knowledge, combine knowledge from different sources, and integrate the information into seq-to-seq architectures. The authors apply their proposal to generative dialog modeling, and apply it to two dialog datasets.

Strengths:
- The paper is well-written and well-motivated.
- The authors evaluate their approach on 2 publicly available datasets and compare it to existing approaches, showing improvements in terms of F1.
The authors conduct a human evaluation to compare their approach against other approaches.

Weaknesses:
- There are some details that are missing from the paper, for example details about the mapping operator, and the specific representations of E_i for the datasets used.
- Parts of the analysis are rushed (e.g., sections 6.2 and 6.3).

Questions/Comments:
- In the human evaluation, is there a difference between the ratings for seen and unseen topics for the Wizard of Wikipedia dataset?
- Section 6.3 (especially the part on the effect of gating) can be improved with additional information/analysis.

**Experience Assessment:**

I have read many papers in this area.

**Review Assessment: Checking Correctness Of Derivations And Theory:**

I assessed the sensibility of the derivations and theory.

**Review Assessment: Checking Correctness Of Experiments:**

I assessed the sensibility of the experiments.

**Review Assessment: Thoroughness In Paper Reading:**

I read the paper at least twice and used my best judgement in assessing the paper.

---

> ### Author Response · Authors · 2019-11-15
> **Thanks for your review!**
>
> Thank you for your review! We have addressed your comments below. Let us know if you have more questions.
>
> Re: details missing (Weakness #1) - We have added a longer description about the mapping operator and representations of E_i for each dataset in Section 4.
>
> Re: rushed analysis (Weakness #2) - We have significantly expanded the analyses in Section 6.2 and Section 6.3 to address your comment. Further, we added additional experiments: we added baselines comparing the scaling effect of our KIF-augmented model to the baseline from Dinan et al (see Figure 3, left) and an ablation study on how the knowledge from difference sources should be combined (Table 5).
>
> Re: human evaluation on seen and unseen topics (Question/Comment #1) - That’s a great question! We added this additional evaluation experiment to Figure 5 (in the Appendix), which shows that the trends from our experiment on the Unseen test set of Wizard of Wikipedia are also reflected in the Seen test set of Wizard of Wikipedia. One interesting difference is on the Unseen test set, the percentage performance over the Retrieval baseline is higher than on the Seen test set. This is because the Retrieval model can only output sentences from the training set as next utterances, so suffers from the topic being excluded in the training set. The KIF-Augmented Transformer can generate from the Transformer decoder, so the model is able to recover more on the Unseen set compared to the Retrieval model.
>
> In addition, beyond what was requested, we re-ran the evaluation on the Unseen test set two more times on different days to understand if there is variance in the evaluation. We added this diagram as Figure 6 (in the Appendix). It shows that for the 5 questions we asked, there was more variance in the “More Interesting” question and “More Human” question, possibly because they are a bit more open-ended than the other questions in the evaluation. In general, the retrieval baseline comparisons seem to have less variance compared to the generative baseline, possibly because the retrieval models are human-written text.
>
> Re: Section 6.3 about Gating (Question/Comment #2) - Thanks, we have improved Section 6.3 by expanding the explanation and analysis. Please let us know if there are additional experiments you would like to see for this ablation. Our goal is to show that not only can the KIF-augmented model incorporate relevant information, but also the KIF-augmented model can ignore irrelevant information via the gating mechanism.

---

### Official Review · AnonReviewer1 · 2019-10-27
**Official Blind Review #1**

**Rating:** 3

**Review:**

This paper uses KNN-based retrieval method for extracting relevant information from different sources for the task of generating response in a dialogue task. They utilized different sources of information from Wikipedia, YFCC image set and dialogue utterances. The method is very straight forward and using nearest neighbors from external information sets as auxiliary information and encoding that information via same neural net to produce the encoded external information. The concatenation of encoded input and auxiliary information is concatenated to produce the output in the dialogue via decoder network. The novelty of the proposed method is incremental over Dinan et al. (2018). The KNN-based retrieval module can produce some drift in dialogue from the actual context of the input which can result in irrelevant response in dialogue. Some failure cases can show the quality changes with those drifts in dialogues.


**Experience Assessment:**

I have published one or two papers in this area.

**Review Assessment: Checking Correctness Of Derivations And Theory:**

I assessed the sensibility of the derivations and theory.

**Review Assessment: Checking Correctness Of Experiments:**

I assessed the sensibility of the experiments.

**Review Assessment: Thoroughness In Paper Reading:**

I read the paper at least twice and used my best judgement in assessing the paper.

---

> ### Author Response · Authors · 2019-11-15
> **Thanks for your review!**
>
> Thanks for your review! We have addressed your comments below.
>
> Re: Novelty- The KIF module we propose is not incremental over previous work - we improve almost 10 F1 points over Dinan et al (2018) (Table 1) and show statistically significant improvements in human evaluation as well (Figure 2). Unlike the attention-based mechanisms to attend to knowledge, KIF scales very well (Figure 3, left). Further, previous work has only focused on one setting for knowledge retrieval (only Wikipedia sentences in Dinan et al 2018) - in contrast, we show that KIF can fetch Wikipedia sentences, dialogue utterances, images, and multiple of these at the same time.
>
> Re: Dialogue Drift- Thanks for raising this important point about dialogue utterance quality. We show F1 gains on both evaluation settings, indicating that our models generate text that is more closely aligned with the human response from the dataset. Further, KIF modules can use the gating mechanism to ignore poor quality knowledge and avoid the dialogue drift issue - in Figure 3 (middle), we explicitly test this with an ablation study where we fetch on a third source of irrelevant knowledge. The KIF-augmented Transformer learns to ignore this source of knowledge, and we see almost no change in performance. Lastly, we see statistically significant improvement compared to the baseline models in human evaluation, even on questions that ask for comparison on “more human” and “more coherent”, indicating a visible improvement in generation quality.
>
>
> We have added several new results:
>
> * We added additional baselines comparing the scaling effect of our KIF-augmented model to the baseline Transformer (see Figure 3, left) - it shows our KIF modules scale much better to fetching on large quantities of knowledge compared to the baseline attention-based models.
> *  We added an ablation study on how the knowledge from difference sources should be combined (Table 5) - it shows that concatenating knowledge from multiple sources works well compared to other forms of combination (such as addition or inner product).
> * We added an additional human evaluation experiment in Appendix Figure 6, which shows that the trends from our experiment on the Unseen test set of Wizard of Wikipedia are also reflected in the Seen test set of Wizard of Wikipedia.
> * We re-ran the human evaluation on the Unseen test set two more times on different days to understand if there is variance in the evaluation. We added this diagram as Appendix Figure 6. For the 5 questions we asked, there was more variance in the “More Interesting” question and “More Human” question, possibly because they are a bit more open-ended than the other questions in the evaluation.
>
> If you have additional experiments or analysis that you would like to see, please let us know.

---

### Official Review · AnonReviewer2 · 2019-10-31
**Official Blind Review #2**

**Rating:** 6

**Review:**

### Summary
​
This paper provides a framework to augment dialogue generation with external data sources using K-Nearest Neighbors in the embedding space. The idea seems simple and intuitive, and the results show improvements over prior work in dialogue generation and retrieval.
​
​
### Strengths
- The paper shows quantitative improvement over some prior works on dialogue agents (however this needs to be correctly validated).
- The reviewer appreciates the human study and conversation example provided in the paper to qualitatively evaluate their model.
- The paper provides ablation studies of the various training tricks for each dataset they have proposed.
​
### Weaknesses
- There are certain changes in the training pipeline that makes the comparison with prior work difficult (Tables 1 and 2), and find the real contribution of data-augmentation caused by KNN-based information fetching (KIF). e.g., In Wizard of Wikipedia experiment, most recent dialogue utterance and turn number are used as salient features. Similarly, the paper utilizes a "personality" feature in ImageChat dataset. It seems the results taken from prior papers have different training settings. Could the authors verify that the extra assumptions made for their model are equivalently applied to other generative baselines? If not, the reviewer recommends to provide some experiments with same conditions.
- Could the authors provide the embedding dimensions and other training details, to assess the contribution of different components of the input's embeddings.
​
​
#### Minor:
- Is there a sound reason behind using concatenation of input representation and fetched representations? This design choice makes the architecture inflexible to the number of data sources. Another way to combine embeddings - e.g. addition or inner product, can also be tried to see if they provide performance improvements.
- It is mentioned that attention based modules scale poorly with large sized datasets. If the authors conducted a quantitative evaluation to test this, it would be valuable to add it to the paper.
- The title and abstract should be limited to generative dialogue modeling, instead of transformers, since the contributions proposed are more suited for this particular application and are quite engineered. Hence, it is not correct to make this claim for transformers in general.
​
​
### Score
Weak Reject (Leaning towards Accept if appropriate experiements can be provided)

**Experience Assessment:**

I do not know much about this area.

**Review Assessment: Checking Correctness Of Derivations And Theory:**

I assessed the sensibility of the derivations and theory.

**Review Assessment: Checking Correctness Of Experiments:**

I assessed the sensibility of the experiments.

**Review Assessment: Thoroughness In Paper Reading:**

I read the paper at least twice and used my best judgement in assessing the paper.

---

> ### Author Response · Authors · 2019-11-15
> **Thanks for your review!**
>
> Thanks for the review and the suggestions. We appreciate the points and have updated the paper to address them. Please let us know if you have additional thoughts about these.
>
> Re: Comparison with prior work (Weakness #1) - For Wizard of Wikipedia, the most recent dialogue utterance and turn number are used as features to improve KIF retrieval of the knowledge, but there is no difference in the dialogue context given to the KIF model and the baseline model. The baseline model can access the last turn and turn number information as it is contained in the dialogue context. We added Appendix Table 6 where a new Wizard of Wikipedia baseline is presented that concatenates the turn information with another copy of the last turn. There is very little improvement over the baseline from Dinan et al (2018), though the turn number is more useful than an additional copy of the last turn.
>
> For the Personality feature in ImageChat- In the Shuster et al (2018) work, the personality feature is embedded into a vector of size 500 to create a personality representation and this is treated as a separate feature. We use the personality to do the KNN fetch (learning a representation of dimension 512) and concatenate it to the dialogue history as a sequence. The dimensionality difference of size 12 is not likely to be relevant. We did add an additional section to Appendix Table 6 where a new Engaging ImageChat baseline is presented that does not have the separate personality feature of Shuster et al (2018) (a no personality baseline) and another experiment that has the personality concatenated to the dialogue history exactly as we use it. There is a performance decrease compared to Shuster et al baseline using our representation, and using no personality is harmful (also observed by Shuster et al).
>
> Note that both Table 1 and Table 2 include re-trained Transformer MemNet baselines on both datasets  (described in Section 5.3). This is because in our work, we used pre-training with Reddit and the pre-trained model was not the same size as the results of the published papers (see Section 5.1 for model sizes). To account for this difference, we re-tuned the baseline architectures to have the same pre-training and model size as our models. For the new baselines we added based on your feedback, they are also consistently use this pre-training and model size.
>
> Re: Additional model details (Weakness #2) - Details such as the embedding dimension, hidden dimension, model size, and learning rate/warmup are present in Section 5.1 “Implementation Details.” We added the following implementation details in our update:
> vocabulary size, dropout, gradient clipping value
> We added a sentence in Section 5.1 about which hyperparameters we tuned as well. Please let us know if this section should be further added to.
>
> Re: Concatenation (Minor #1) - Good suggestion, that’s very interesting. We added a Appendix Table 5 with an ablation study on how to combine the input representation with the fetched representations. We found on 2 data sources on Wizard of Wikipedia, the both addition and inner product had worse performance. In our 3 data source setting where we test if the model can ignore an irrelevant knowledge source (Figure 3 middle), the inner product combination performed better, though addition was still not so good. It is possible if we extend to even more data sources, inner product could be better.
>
> Re: Attention Modules Scale Worse (Minor #2) - Thanks for this suggestion, it is definitely important to highlight the advantage of our model scaling-wise. We updated Figure 3 (left) to compare the ability of KIF v. attention based mechanism (from Dinan et al (2018)) on scaling to fetching information from not just the TF-IDF output created by Dinan et al that averages 34 sentences, but on the entire Wikipedia article as well as multiple articles from related topics. Originally we only showed the scaling ability of KIF, but we added experiments now with the Generative Transformer MemNet that demonstrate the advantage of KIF.
>
> Re: Title/Abstract on Transformers (Minor #3) - Thanks, we have adjusted the abstract to reflect that we are augmenting sequence-to-sequence Transformer architectures applied to generative dialogue tasks.
>
> We did additional human evaluation experiments: Appendix Figure 5 shows that the trends on the Unseen test set of Wizard of Wikipedia are also reflected in the Seen test set. In addition, beyond what was requested, we re-ran the evaluation on the Unseen test set two more times on different days to understand if there is variance in the evaluation. We added this diagram as Figure 6. It shows that for the 5 questions we asked, there was more variance in “More Interesting” and “More Human”, possibly because they are a bit more open-ended. In general, the retrieval baseline comparisons seem to have less variance compared to the generative baseline, possibly because the retrieval models are human-written text.

---

### Decision · Program_Chairs · 2019-12-19

**Decision:**

Reject

**Comment:**

This paper augments transformer encoder-decoder networks architecture with k nearest neighbors to fetch knowledge or information related to the previous conversation, and demonstrates improvements through manual and automated evaluation. Reviewers note the fact that the approach is simple and clean and results in significant improvements, however, the approach is incremental over the previous work (including https://arxiv.org/pdf/1708.07863.pdf). Furthermore, although the authors improved the article in the light of reviewer suggestions (i.e., rushed analysis, not so clear descriptions) and some reviewers increased their scores, none of them actually marked the paper as an accept or a strong accept.